# How big does your neural network have to be?: A Scaling Law Study in Multi-Spectral Remote Sensing

## Abstract

Remote sensing imagery from systems such as Sentinel provides full coverage of the Earth's surface at around 10 meter resolution. The remote sensing community has transitioned to extensive use of deep learning models based on their high performance on benchmarks such as the ISPRS Vaihingen. Convolutional models such as UNet and ResNet variations are commonly employed for remote sensing but typically only accept three channels due to their development for RGB imagery, while Sentinel satellite systems have more than 10. Recently, a number of transformer architectures have also been proposed for remote sensing, but they typically have not been extensively benchmarked and have only been employed on rather small datasets. Meanwhile, it is becoming possible to obtain dense spatial land-use labels for entire first-level administrative divisions of some countries. Scaling law observations indicate that substantially larger, multi-spectral transformer models may provide a huge leap in the performance of remote sensing models in these settings. In this work, we develop a family of multi-spectral transformer models, which we evaluate across orders of magnitude differences in model parameters to evaluate their performance and scaling effectiveness on a densely labeled imagery dataset. We develop a novel multi-spectral attention strategy and demonstrate its effectiveness through ablations. We further show in this setting that models many orders of magnitude larger than conventional architectures such as UNet lead to substantial improvements in accuracy: a UNet++ model with 23M parameters results in less than 65% accuracy, while a multi-spectral transformer with 655M parameters yields an accuracy of over 95% on the Biological Valuation Map of Flanders. A link to open source code will be provided in the camera ready document.

## 1 Introduction

Remote sensing plays a crucial role in environmental monitoring, urban planning, disaster forecasting, and more by utilizing rich data from satellite and aerial systems. Processing this vast, high-dimensional data poses significant challenges, especially with traditional, time-consuming, and error-prone techniques. Recent advances in machine learning (ML), particularly convolutional neural networks (CNNs), have greatly enhanced the accuracy and efficiency of remote sensing analysis by automatically learning complex spatial and spectral patterns (Zhang et al., 2016; Li et al., 2018).

CNNs, originally developed for image tasks like classification, detection, and segmentation, have been effectively introduced into remote sensing. For instance, Hu et al. (2015) demonstrated the effectiveness of CNNs in hyperspectral image classification, achieving state-of-the-art performance at that time. CNNs have since been applied to land-use classification, terrain change detection, and urban planning.

Transformers, initially successful in natural language processing (Vaswani, 2017), have been adapted for vision tasks, improving the handling of distant dependencies and complex spatial correlations via attention modules (Carion et al., 2020; Han et al., 2022). Dosovitskiy (2020) introduced the Vision Transformer (ViT), which outperforms CNN-based models on various image classification benchmarks, suggesting its potential for remote sensing tasks. Liu et al. (2021) designed shifted

window transformers (Swin-T), providing an energy-efficient variation of transformer. He et al. (2022) later demonstrated their superiority over conventional CNNs in multi-modal remote sensing data. These works highlight the importance of architectural choices in neural networks for remote sensing.

Despite these advancements, the scaling properties of neural networks in remote sensing remain under-explored. While larger models and datasets might intuitively lead to better performance, the relationship between model size, data size, and task performance in remote sensing contexts has not been thoroughly investigated. Exploring these scaling laws (Kaplan et al., 2020) is necessary to develop efficient and accurate models.

In this paper, we first introduce the current state of research in remote sensing machine learning and point out the model and dataset mismatching issues that mainstream machine learning applications currently face. Then we propose a spatial-spectral module fused Swin Transformer backbone for multi-spectral segmentation tasks. We systematically explore the scaling laws in remote sensing and demonstrate that our backbone achieves state-of-the-art performance by conducting an extensive empirical study on a selection of neural network architectures. Furthermore, we investigate how performance varies with model size, architecture selection, and training data size. Additionally, we perform an ablation study to examine the impact of different components on the scaling behaviors of exemplary simple architectures. Our results provide meaningful insights on how to scale neural network models according to specific application needs in multi-spectral remote sensing tasks.

## 2 BACKGROUND AND MOTIVATION

### 2.1 MACHINE LEARNING IN REMOTE SENSING

Machine learning (ML) has significantly changed the remote sensing research with deep learning methods, convolutional neural networks (CNNs) are growing to be central to processing and analyzing complex remote sensing data. Early applications of ML in remote sensing focused on using traditional algorithms, such as support vector machines and decision trees, for simple tasks like land cover classification and change detection. As more accurate and powerful tools and architecture design choices come out in recent years, the implementation of CNNs has led to substantial performance improvements in various complicated tasks such as image classification, object detection, and semantic segmentation (Zhu et al., 2017). CNNs also showed effectiveness in processing hyperspectral images and significantly outperformed traditional methods by learning hierarchical feature representations directly from data (Hu et al., 2015). For high-resolution satellite imageries, Maggiori et al. (2016) has proven that tailor-made deep CNNs can handle large-scale remote sensing image classification and achieve state-of-the-art accuracies.

Transformer models have further reshaped and accelerated the advances in remote sensing research with even better accuracy and abilities to handle complex input samples. Originally designed for natural language processing tasks, transformers were later adapted to vision tasks as Vision Transformers (ViT) by Dosovitskiy (2020), outperforming CNNs in some tasks by leveraging long-range spatial dependencies with attention mechanisms. Because of its superior performance on image classification tasks, ViT showed potential applicability to remote sensing tasks. Their work was later extended by applying transformers to multi-modal remote sensing data, demonstrating architectural design choices are crucial for model performance across different data modalities and scales (Aleissaee et al., 2023).

Enlightened by the spatial attention mechanism in Transformer architectures, researchers have been trying to adapt this mechanism for multi/hyper-spectral remote sensing tasks actively. Hang et al. (2021) introduced spectral-attention-aided CNN models and proved that it is beneficial to include spectral-attention modules into CNN backbones. Later work of Roy et al. (2021) demonstrated that ResNet can be improved by incorporating spectral attention modules. Moreover, Zhong et al. (2022) designed an spectral association block that focuses on establishing the connection between different locations in the cuboid by calculating spectral association kernels using 3D convolutions. Although spectral information is included in the training process, their methods are all specified in sparse representations of spectral correlation but not learning from token-based to spectral band-based inter-channel attention.

## 2.2 DATASET SIZE DOMINATES MODEL SELECTION

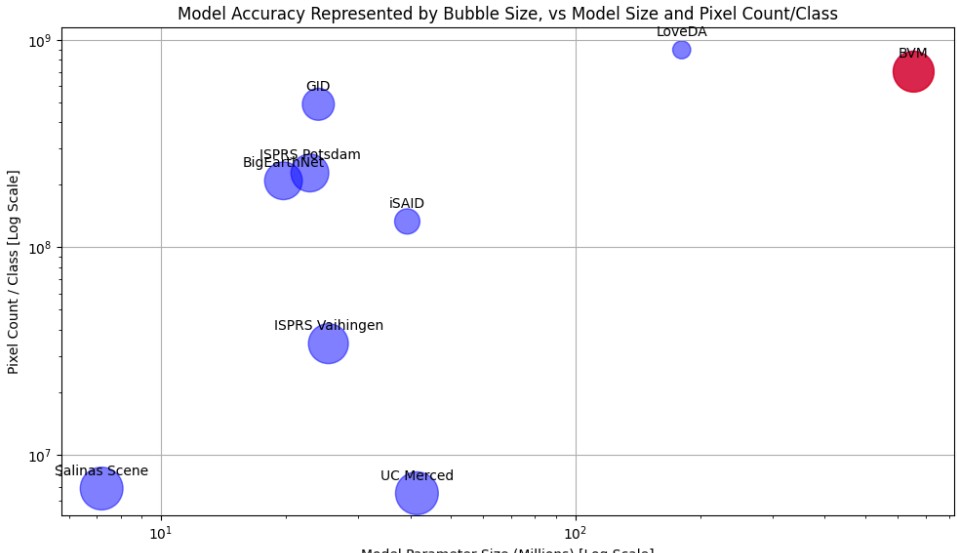

Figure 1: Comparison of the size of mainstream remote sensing datasets and the BVM dataset (Li et al., 2024a) (dataset size vs. model size). Y-axis represents the overall dataset size in the form of total pixel number per class in each dataset. X-axis is the averaged parameter size of models used on each of the datasets. The size of bubbles represents averaged accuracy scores. The BVM dataset is placed at the top-right corner, indicating both the model size and dataset size are well above the current norms of the latest research.

Remote sensing and machine learning researchers seem to have been stranded by the ideas of developing novel adaptations of existing deep learning architectures and sophisticated numerical methods to improve the performance of individual models on individual datasets (Bergamasco et al., 2023; Peng et al., 2023; Roy et al., 2023; Yuan et al., 2023; Lv et al., 2023; Zhang et al., 2023). High-accuracy learning results are often viewed as excellent where specific settings of models are tuned for a specific task. Few have studied the scaling properties of models that are used, especially in the remote sensing field where more and more very large datasets are merging out. It is important to point out that in the current state of remote sensing research, most models being employed are relatively small, typically containing fewer than 200 million parameters. Taking ResNet50, for example, with a small size of 25 million parameters, it has been actively applied to datasets ranging from tiny ones like Salinas Scenes (Graña et al., 2021), to very large datasets such as BigEarthNet (Clasen et al., 2024). Even after the adaptation of large vision transformers to remote sensing, these models are still applied to datasets of varying scales without considering the efficiency and performance gains. It became necessary to do empirical studies on a very large dataset with spatial-spectral models of different sizes to advocate a paradigm that matching the scaling properties of models with dataset size is crucial for benchmarking models properly.

In this paper, we use the Biological Valuation Map (BVM) of Flanders (Li et al., 2024a), a recently published multi-spectral land cover dataset with pixel-wise labeling consisting of Sentinel-2 public imagery, as the training dataset. The scale of the experiment positions our work in a different region of the research field, as seen in Figure 1 and Appendix A.1, where larger models and larger datasets are needed to push the boundaries of performance instead of sticking to small datasets and small model studies. For example, datasets such as UCMerced (with 137M pixels) and ISPRS Vaihingen (with around 169M pixels) are widely used for benchmarking models of which parameter size could range from but are considerably smaller than BVM dataset in both pixel count and complexity. Similarly, the ISPRS Potsdam dataset contains roughly 1.37 billion pixels, which, while larger than Vaihingen, is still much smaller than the scale we are dealing with (subsection A.1). There are also larger satellite datasets like LoveDA and BigEarthNet, but their labels are either unreliable or sparse. Our research operates on a different scale, focusing on advancing large-scale segmentation tasks.

### 2.3 SCALING ARCHITECTURES ON MULTI-SPECTRAL REMOTE SENSING TASKS

Scaling laws demonstrate that increasing model size, data volume, and computing improves neural network performance, as first formalized by Kaplan et al. (2020) and extended to generative models (Henighan et al., 2020) and transfer learning (Hernandez et al., 2021). The CNN model architecture can greatly affect its effectiveness in remote sensing tasks; determining hyperparameters such as the depth, width, and input dimensions can often be up to the choices of engineers. EfficientNet introduced by Tan (2019) pioneered architectural scaling by introducing a compound scaling method with a fixed set of hyperparameter coefficients. This approach yielded state-of-the-art results on multiple benchmarks with fewer parameters and reduced computational cost.

Meanwhile, Rosenfeld et al. (2021) examines how the performance of pruned neural networks scales with model size, data volume, and compute resources, highlighting the importance of matching model complexity with data to prevent overfitting and inefficient computation. Zhang et al. (2022) further explored the relationship between architectural design and scaling laws, confirming that deeper and wider models tend to benefit more than shallow and narrow ones, suggesting the importance of architectural design in optimizing model performances. Although scaling laws of vision transformers have been studied (Zhai et al., 2022), the problem in multi-spectral vision tasks could be influenced by the composition of different modules in models. Our study fills the gap between their holistic analysis and multi-spectral cases.

## 3 METHODOLOGIES

In this section, we first introduce our spectral dependency module. Then we show our SDM-infused scaling-efficient Swin transformer backbone that was adapted for large-scale multi-spectral training. We define a scaling factor that provides a quantitative approach to measure the scalability of architectures.

### 3.1 SPECTRAL DEPENDENCY MODULE

We introduce a Spectral Dependency Module (SDM), whose design is similar to the spatial attention head in the Transformer, modifying the traditional attention mechanism by replacing token-based attention with spectral band-based attention. Instead of treating tokens as the basic unit of input, the SDM module considers spectral bands in multi-spectral data as the fundamental elements.

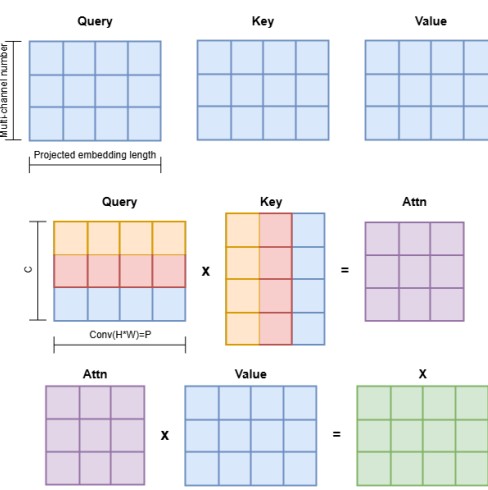

Figure 2: SDM block mechanism, similar to transformer attention head, the query token number dimension is substituted by multi-spectral channel number. Embeddings from same channels are marked in the same color.

Each spectral band is rearranged into patches, and these patches of each channel are projected into an embedding space, similar to how tokens are projected in classical attention mechanisms. The core of this module is to compute the channel dependencies between all spectral bands across different channels. By computing attention weights based on spectral dependencies, the SDM allows for a fine-grained representation of spectral relationships, enabling improved performance in tasks requiring multi-spectral data processing.

Let $X \in \mathbb{R}^{C \times H \times W}$ denote the multi-spectral data, where:

- $C$ is the number of spectral bands (channels),
- $H$ and $W$ are the spatial dimensions (height and width).

For each spectral band $c \in \{1, 2, \ldots, C\}$, we rearrange its data $X_c \in \mathbb{R}^{H \times W}$ into a set of patches. The patches are represented as:

$$\mathbf{x}_{c,p} \in \mathbb{R}^P, \quad p \in \{1, 2, \ldots, N_p\}.$$

Each patch $\mathbf{x}_{c,p}$ is projected into an embedding space using a linear projection function $E : \mathbb{R}^P \to \mathbb{R}^d$:

$$\mathbf{z}_{c,p} = E(\mathbf{x}_{c,p}) \in \mathbb{R}^d.$$

For each patch index $p$, we stack the embeddings of all spectral bands to form a matrix:

$$\mathbf{Z}_p = \begin{bmatrix} \mathbf{z}_{1,p} \\ \mathbf{z}_{2,p} \\ \vdots \\ \mathbf{z}_{C,p} \end{bmatrix} \in \mathbb{R}^{C \times d}.$$

We compute the query ($\mathbf{Q}_p$), key ($\mathbf{K}_p$), and value ($\mathbf{V}_p$) matrices for each patch $p$ using learnable weight matrices $\mathbf{W}_Q$, $\mathbf{W}_K$, and $\mathbf{W}_V$:

$$\mathbf{Q}_p = \mathbf{Z}_p \mathbf{W}_Q \in \mathbb{R}^{C \times d_k},$$
$$\mathbf{K}_p = \mathbf{Z}_p \mathbf{W}_K \in \mathbb{R}^{C \times d_k},$$
$$\mathbf{V}_p = \mathbf{Z}_p \mathbf{W}_V \in \mathbb{R}^{C \times d_v}.$$

The correlation weights are calculated by computing the scaled dot-product attention over the spectral bands for each patch:

$$\mathbf{A}_p = \text{softmax}\left( \frac{\mathbf{Q}_p \mathbf{K}_p^\top}{\sqrt{d_k}} \right) \in \mathbb{R}^{C \times C}.$$

The softmax function is applied row-wise to ensure that the attention weights for each spectral band sum to 1. By using the attention weights $\mathbf{A}_p$, we compute the output embeddings for each patch:

$$\mathbf{O}_p = \mathbf{A}_p \mathbf{V}_p \in \mathbb{R}^{C \times d_v}.$$

The outputs $\mathbf{O}_p$ from all patches are aggregated by averaging to form the final representation used for downstream tasks:

$$\mathbf{O} = \text{Avg}\left( \{\mathbf{O}_p\}_{p=1}^{N_p} \right).$$

SDM combined with classical vision transformer attention blocks can capture both spatial and inter-channel dependencies, thus enhancing spectral-spatial feature learning. It also introduces a scheme for building and scaling hybrid models by combining SDM blocks with classical attention heads and convolutional layers. This modular approach enables flexible model scaling by adjusting the number and configuration of SDM, attention, and convolution blocks. Additional specialized blocks could also be integrated into this structured scaling framework, allowing for customizable architectures tailored to different data types and tasks.

## 3.2 Architectures and Multi-Spectral Swin Transformer

In this paper, we selected UNet++ (Zhou et al., 2018), ResNet (He et al., 2016), and Swin Transformer (Liu et al., 2021) for our scaling laws study and our customized spatial-spectral module blocks for modular scaling behavioral study, due to their simplicity and scalability. These models scale easily—ResNet by adding residual blocks, Transformers through layers or embedding dimensions, and our customized model via its modular design—allowing for detailed studies of scaling laws. Moreover, each model demonstrates consistent performance improvements with increasing scale, as seen with deeper ResNets (He et al., 2016), larger transformers (Brown, 2020) and in our experiments.

In this paper, we modified the Swin transformer models with SDM by connecting it to the multi-head self-attention and shift-window multi-head self-attention modules in each transformer block at stage 1. As Figure 3 shows,

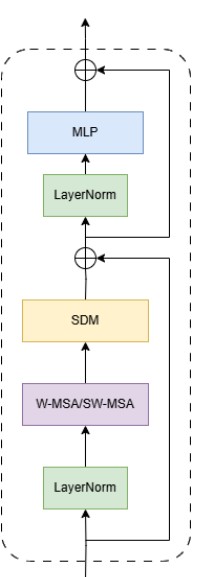

Figure 3: The transformer block of a MS-Swin model

the output dimension of MSA modules is the same as SDM's input dimensions, so it allows seamless connection of two modules. Our choice to insert SDM at stage 1 is due to the consideration that the first stage preserves the most raw spectral features that the SDM module may need. We refer to this modified architecture as Multi-Spectral Swin Transformer (MS-Swin).

### 3.3 EXPERIMENT SPECIFICATIONS

In this paper, instead of using an end-to-end training strategy, we choose to train the model with griding and patching (Chen et al., 2014) by first griding the large training images into smaller cells, then randomly sampling a certain number of cells to form up patches. This approach not only makes it feasible to train on high-resolution images by lowering memory requirements but also increases the diversity of training data, as multiple patches are extracted from a single image. By focusing on local patterns within these patches or grids, the strategy enhances the model's ability to learn fine-grained spatial features and improves overall generalization performance (Li et al., 2016). We use Overall Accuracy (OA) as our metric for model performance because it provides a straightforward, global measure of how well the model correctly classifies pixels across all classes.

The scaling efficiency coefficient $S$ quantifies how effectively a neural network scales its performance relative to the increase in parameters and computational resources. It is mathematically defined as:

$$S = -\frac{1}{\log\left(\frac{G}{P \times C}\right)}$$

where:

- $G$ is the Performance Gain Factor.
- $P$ is the Parameter Count Scaling Factor.
- $C$ is the Computation Increase Factor.

The proposed coefficient provides a quantitative measure of scaling efficiency for neural networks. By considering both the performance gain and the increase in the consumption of computational resources (parameter count and computation increase), the coefficient evaluates how effectively a model scales. A higher coefficient indicates that a model achieves a larger performance improvement with relatively modest increases in parameters and training time, making it more efficient in scaling. Conversely, a lower coefficient suggests that either the scaling process is inefficient or the model is approaching the limit of diminishing returns. At this stage scaling the model further with significant resource investments results in only marginal performance gains.

## 4 RESULTS AND DISCUSSION

For all our experiments, we utilized a single-node setup equipped with four NVIDIA A100-80G GPUs. We adopted distributed data parallelism with shared gradient synchronization across the GPUs, and the gradient accumulation technique was managed through the HuggingFace Accelerate API, which allowed efficient distribution of the training load while reducing the memory requirements by accumulating gradients over multiple mini-batches before updating the model weights.

For the optimizer, we employed the Adam optimizer because of its adaptive learning rate properties, making it well-suited for large-scale training tasks. All GPUs were contained within a single node to make sure training processes introduced no inter-node communications, and all input samples were treated in read-only mode, minimizing the possible I/O overhead. We reserve sufficient memory space regardless of the model size to keep our training settings constant and stable for the ablation study.

As evident by Figure 4 and Table 1, the Swin and MS-Swin transformer models exhibit superior performance compared to both the ResNet family and the UNet++ model across all sizes, based on the provided data. In the small model category, Swint (27 million parameters) achieves an accuracy of 91.34%, significantly outperforming ResNet-20M (20 million parameters) with 80.95% accuracy and UNet++ (23 million parameters) with 64.48% accuracy. This substantial accuracy gap highlights the efficiency of Swin models in handling complex tasks with fewer parameters.

Table 1: Comparison of models with parameters, training time, accuracy, and scaling coefficients.

| Model | Parameters (M) | Time/Epoch (h) | Accuracy (%) | Scaling Coefficient |
|---|---|---|---|---|
| **Small Models ( 1M to 30M Parameters)** | | | | |
| ResNet-1M | 1 | 3.5 | 75.92 | Baseline |
| ResNet-2M | 2 | 3.9 | 76.03 | 2.879 |
| UNet++ | 23 | 4.0 | 64.48 | N/A |
| ResNet-20M | 20 | 4.1 | 80.95 | 0.745 |
| Swint | 27 | 8.7 | 91.34 | Baseline |
| MS-Swint | 27 | 8.7 | 92.25 | Baseline |
| **Medium Models ( 50M to 100M Parameters)** | | | | |
| ResNet-230M | 230 | 7.4 | 84.10 | 0.378 |
| Swins | 49 | 12.6 | 92.19 | 2.406 |
| MS-Swins | 49 | 12.6 | 92.53 | 2.390 |
| Swinb | 86 | 14.8 | 93.08 | 1.378 |
| MS-Swinb | 86 | 14.8 | 93.38 | 1.373 |
| **Large Models ( 150M to 300M Parameters)** | | | | |
| ResNet-1550M | 1550 | 25.2 | 87.32 | 0.251 |
| Swinl | 195 | 16.3 | 94.57 | 0.896 |
| MS-Swinl | 195 | 16.3 | 94.80 | 0.893 |
| **Extra-Large Models ( 650M to 2800M Parameters)** | | | | |
| ResNet-2800M | 2800 | 40 | 89.19 | 0.225 |
| Swinh | 655 | 24.0 | 96.64 | 0.555 |
| MS-Swinh | 656 | 24.0 | 96.71 | 0.554 |

Additionally, Swin models maintain higher scaling coefficients than ResNet models, indicating more efficient scaling as model size increases. The hierarchical architecture and shifted window mechanism of Swin transformers contribute to their enhanced performance and scalability, making them favorable choices over traditional convolutional models like ResNet and UNet++.

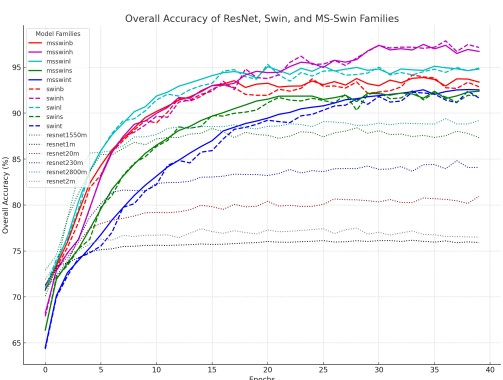

Figure 4: Validation accuracy curves of scaled networks, "m" in legend stands for million parameters, and "epochs" stands for how many times the entire dataset has passed through the model. Curve for UNet++ can be found in appendix

Between the Swin and MS-Swin models, the MS-Swin models offer additional advantages, making them a better choice for multi-spectral segmentation tasks. The integration of the Spectral Dependency Module (SDM) into the Swin Transformer architecture allows MS-Swin models to capture spectral dependencies more effectively, leading to higher accuracies without a significant increase in parameters or loss of scaling efficiency. For example, MS-Swint achieves an accuracy of 92.25%, approximately 0.91% higher than Swint, with almost identical scaling coefficients. This demonstrates that MS-Swin models enhance spectral feature learning while maintaining efficient scalability. Therefore, the MS-Swin models are optimal for multi-spectral tasks requiring high performance and scalability, as they provide superior accuracy and maintain scaling efficiency compared to both their Swin counterparts and conventional models like ResNet and UNet++.

Figure 5 shows the loss curve comparison between MS-Swin and Resnet. The analysis of loss curves for various neural network models reveals distinct behaviors based on model size and architecture. Smaller ResNet models like ResNet1m and ResNet20m show rapid early loss descent but quickly reach saturation, indicating a capacity limitation in handling complex patterns as training progresses. In contrast, larger ResNet models and all MS-Swin Transformer models demonstrate a more gradual loss reduction, suggesting that they can continue improving with extended training due to their greater capacity to learn complex data representations.

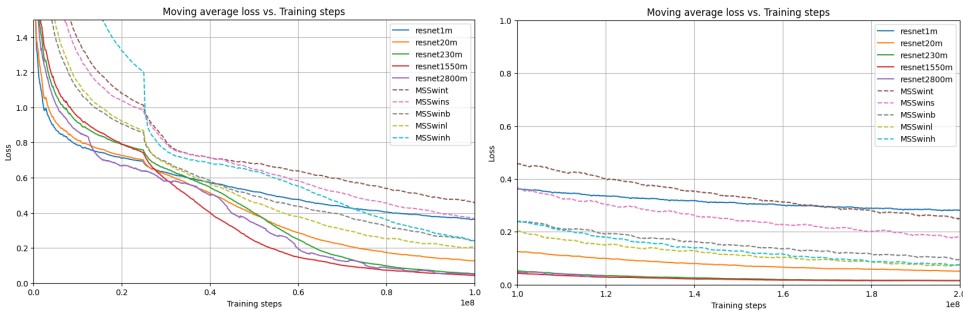

Figure 5: Moving average loss curves of scaled networks, left: initial region of training, right: stable descending region

MS-Swin Transformer models exhibit a slower but more persistent decline in loss, suggesting that they benefit from extended training epochs before showing signs of saturation, unlike conventional models like ResNet. This prolonged effectiveness in learning indicates that MS-Swin Transformers, with their MSA and SDM modules, are capable of exploiting their architectural efficiency to handle complex data relationships over longer periods. While larger ResNet models tend to plateau earlier, larger MS-Swin models, like MS-Swinl and MS-Swinh, continue to show potential for improvement well beyond the typical saturation points of conventional architectures, highlighting the distinct advantage of transformer-based models in sustained learning capability. We can also observe the scaling stops being rewarding for ResNet at around 1e8 sample passes for the loss gap between ResNet1550m and ResNet2800m is not distinct anymore while the loss MS-Swinl and MS-Swinh only start to converge at around 2e8 sample passes. This behavior underscores the fact that transformer models, despite their size, are inherently designed to scale more effectively with increasing data and training duration before experiencing diminishing returns.

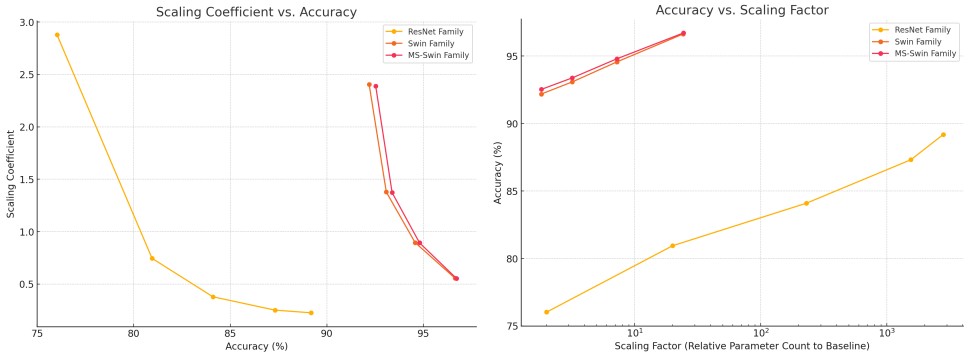

Figure 6: Scaling properties of two architectural families. The "scaling coefficient" indicates how efficiently the network architecture scales. "Scaling factor" represents how many times the current model is larger in size compared to the baseline model. A higher scaling coefficient implies better scalability of the model, meaning the model does not suffer significantly from diminishing returns at that scaling step.

Figure 6 shows comparative analysis of MS-Swin, Swin, and ResNet model families. MS-Swin models not only consistently maintain higher accuracy across all scaling scales but also demonstrate superior scalability. Both MS-Swin and Swin showed much higher scaling efficiency than ResNet at the same accuracy level, and the MS-Swin model has even higher scaling efficiency than the Swin model. This indicates that MS-Swin only suffers from diminishing return at very high accuracy levels and at very large scales. Although data on larger MS-Swin models is limited due to hardware constraints, it is projected that their scaling properties would outperform those of ResNet.

## 4.1 MODEL SIZE SHOULD MATCH DATASET SCALE

As Table 3 and Figure 7 show, when exposed to a small amount of data, small models, which have fewer parameters, are more appropriate for scenarios with limited data. The small models are significantly more energy efficient than their larger counterparts, with virtually the same level of accuracy scores. These models tend to generalize better and are less likely to overfit when data is scarce. As the dataset size grows, larger models begin to show their advantages. With more data, these models can learn more complex patterns, which is evident from the performance jump observed in large and extra-large models with increased training data.

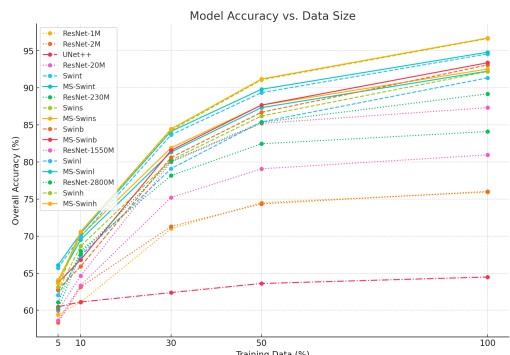

Figure 7: The accuracy curves of models vs. training data size.

The impact of increasing model size becomes more obvious with larger datasets. Smaller models reach their capacity earlier, while larger models continue to improve. We can also infer from the Figure 7 that there is still room for further increase in MS-Swin and Swin models, demonstrating that at this dataset size scale, it is significantly more important to scale up model size than to optimize the architectural design of the network since even conventional architectural choices such as ResNet can perform much better than using smaller variations.

## 5 CONCLUSIONS AND LIMITATIONS

In this paper, we have explored the scaling laws governing neural networks applied to large-scale, multi-spectral remote sensing tasks. Our extensive experiments on the Biological Valuation Map (BVM) of Flanders—a densely labeled, large-scale dataset—demonstrated that larger models with appropriate architectural choices significantly outperform traditional architectures like UNet++ and ResNet. By introducing a Spectral Dependency Module (SDM) integrated into a Swin Transformer architecture, we developed a Multi-Spectral Swin Transformer (MS-Swin) model that effectively captures spectral dependencies in multi-spectral data. Specifically, we showed that our MS-Swin models achieve superior accuracy and scaling efficiency, with models containing hundreds of millions of parameters yielding over 96% accuracy, while smaller conventional models lag behind. These findings underscore the importance of matching model complexity to dataset scale and leveraging advanced architectural designs to harness the full potential of large multi-spectral datasets.

Despite the promising results, our study has several limitations. Firstly, our experiments were conducted solely on the BVM dataset, which is restricted to the Belgian Flemish region. To generalize our findings, it is necessary to test the proposed models on a broader range of land regions in different countries, leveraging multi-spectral datasets of similar scale, which governments and organizations are increasingly capable of providing. Secondly, due to computational resource constraints, we were unable to further scale up the size of the MS-Swin models. Future work could explore even larger models, potentially requiring inter-node training setups, although this would introduce additional computational overhead that must be carefully managed. Lastly, while we focused on comparing our models with standard architectures like ResNet and UNet++, there exists a wide variety of state-of-the-art models, including other large transformers and conventional models. Incorporating and testing the Spectral Dependency Module within these architectures could provide deeper insights into their scaling behaviors and further consolidate the configurations of SDM.

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

# A APPENDIX

## A.1 DATASET SCALE

Here we attach additional information for an overview of some widely-adopted remote sensing datasets and compare their scales and related models:

Table 2: A comparison of mainstream datasets and BVM dataset, and the models applied to them. Note that the pixel count is an estimation based on dataset specifications, and not all datasets and models are covered.

| Dataset | Pixel Count | Classes | Model | Accuracy | Reference |
|---|---|---|---|---|---|
| Salinas Scene | 0.11M | 16 | PSE-UNet | 91.01% (OA) | (Li et al., 2022) |
| | | | 3D-CNN | 97.55% (OA) | (Liu et al., 2023) |
| | | | HybridSN | 99.84% (OA) | (Roy et al., 2020) |
| | | | SMALE | 99.28% (OA) | (Liao et al., 2024) |
| UC Merced | 137M | 21 | DenseNet-121 | 99.88% (OA) | (Bi et al., 2022) |
| | | | MS2AP | 99.01% (OA) | (Bi et al., 2021b) |
| | | | LSENet | 98.69% (OA) | (Bi et al., 2021a) |
| | | | VGG-VD16 | 95.21% (OA) | (Özyurt et al., 2020) |
| ISPRS Vaihingen | 206M | 6 | PGNet | 86.32% (OA) | (Liu et al., 2022) |
| | | | MANet | 86.51% (OA) | (Li et al., 2021a) |
| | | | EMNet | 95.42% (OA) | (Li et al., 2023a) |
| | | | DeepLabv3+ | 86.07% (OA) | (Chen et al., 2018) |
| ISPRS Potsdam | 1.37B | 6 | CM-UNet | 91.86% (OA) | (Liu et al., 2024) |
| | | | SSCNet | 91.03% (OA) | (Li et al., 2023b) |
| | | | HCANet | 90.15% (OA) | (Li et al., 2021b) |
| | | | AerialFormer-B | 91.4% (OA) | (Yamazaki et al., 2023) |
| | | | DC-Swin | 92% (OA) | (Wang et al., 2022a) |
| iSAID | 2B | 15 | SegNeXt-L | 70.3% (IoU) | (Guo et al., 2022) |
| | | | SegNeXt-B | 69.9% (IoU) | (Guo et al., 2022) |
| | | | AerialFormer-B | 69.3% (IoU) | (Yamazaki et al., 2023) |
| LoveDA | 6.27B | 7 | UNet-Ensemble | 56.16% (IoU) | (Dimitrovski et al., 2024) |
| | | | SFA-Net | 54.9% (IoU) | (Hwang et al., 2024) |
| | | | ViT-G12X4 | 54.4% (IoU) | (Cha et al., 2023) |
| GID | 7.34B | 15 | LSKNet-S | 82.3% (OA) | (Li et al., 2024b) |
| | | | DeepTriNet | 77% (OA) | (Ovi et al., 2023) |
| BigEarthNet | 9B | 43 | ResNet50 | 91.8% (OA) | (Wang et al., 2023) |
| | | | ViT-S | 89.9% (OA) | (Wang et al., 2023) |
| | | | ResNet18 | 89.3% (OA) | (Wang et al., 2022b) |
| **BVM** | **10.57B** | **15** | **Swin-h** | **96.71% (OA)** | (Li et al., 2024a) |

## A.2 STATISTICS ON PART OF TRAINING DATA

Table 3: Overall accuracy (%) of models trained on different percentages of the training dataset. All models were validated on the full test set.

| Model | 5% Data | 10% Data | 30% Data | 50% Data | 100% Data |
|---|---|---|---|---|---|
| **Small Models (˜1M to 30M Parameters)** | | | | | |
| ResNet-1M | 59.40 | 61.07 | 71.00 | 74.52 | 75.92 |
| ResNet-2M | 58.33 | 63.14 | 71.30 | 74.35 | 76.03 |
| UNet++ | 60.47 | 61.11 | 62.38 | 63.61 | 64.48 |
| ResNet-20M | 58.61 | 63.35 | 75.20 | 79.08 | 80.95 |
| Swint | 62.05 | 67.35 | 79.10 | 85.40 | 91.34 |
| MS-Swint | 64.02 | 69.51 | 81.33 | 87.33 | 92.25 |
| **Medium Models (˜50M to 100M Parameters)** | | | | | |
| ResNet-230M | 61.08 | 67.99 | 78.16 | 82.45 | 84.10 |
| Swins | 62.65 | 68.65 | 80.25 | 86.20 | 92.19 |
| MS-Swins | 64.06 | 70.02 | 81.91 | 87.66 | 92.53 |
| Swinb | 62.80 | 65.90 | 80.60 | 86.70 | 93.08 |
| MS-Swinb | 63.71 | 66.80 | 81.56 | 87.67 | 93.38 |
| **Large Models (˜150M to 300M Parameters)** | | | | | |
| ResNet-1550M | 59.95 | 64.65 | 79.99 | 85.20 | 87.32 |
| Swinl | 65.70 | 69.90 | 83.65 | 89.35 | 94.57 |
| MS-Swinl | 66.09 | 70.36 | 84.13 | 89.80 | 94.80 |
| **Extra-Large Models (˜650M to 2800M Parameters)** | | | | | |
| ResNet-2800M | 60.19 | 67.59 | 80.08 | 85.34 | 89.19 |
| Swinh | 63.12 | 70.53 | 84.27 | 91.08 | 96.64 |
| MS-Swinh | 63.66 | 70.60 | 84.45 | 91.19 | 96.71 |

## A.3 COMPLETE ACCURACY CURVES

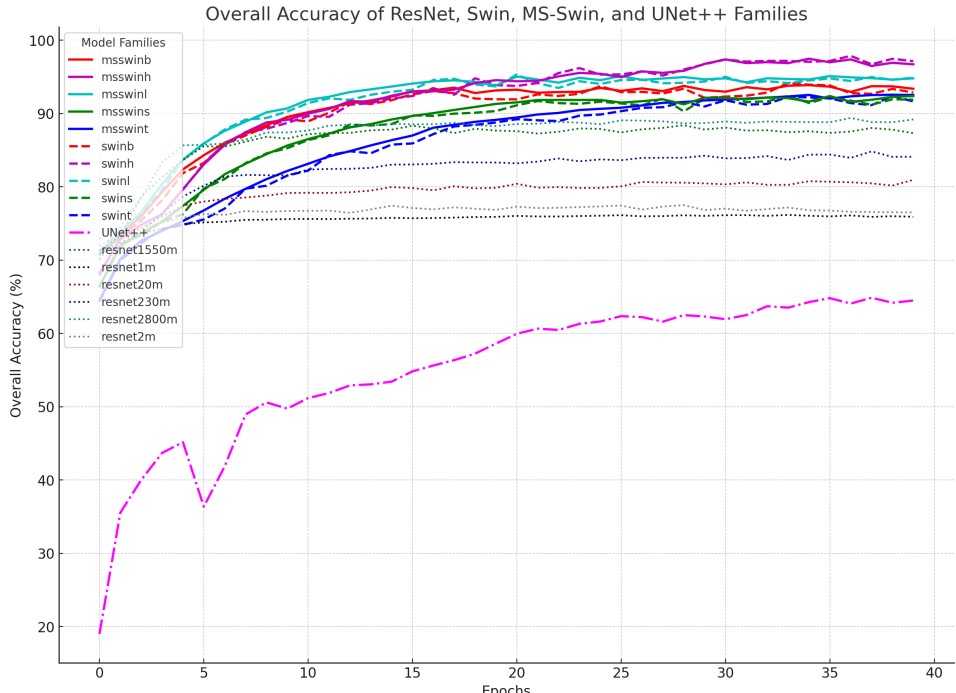

Figure 8: Validation accuracy curves of scaled neural networks, including UNet++ model.

## A.4 ABLATION STUDY ON SCALING PROPERTIES OF DIFFERENT COMPONENTS

We further demonstrate the impact of convolutional, MSA and SDM module's impact on scaling behaviors of neural network models through testing their different combinations. We simply build the models by connecting 2D-convolutional layer, MSA and SDM in series, and append a classification head to it.

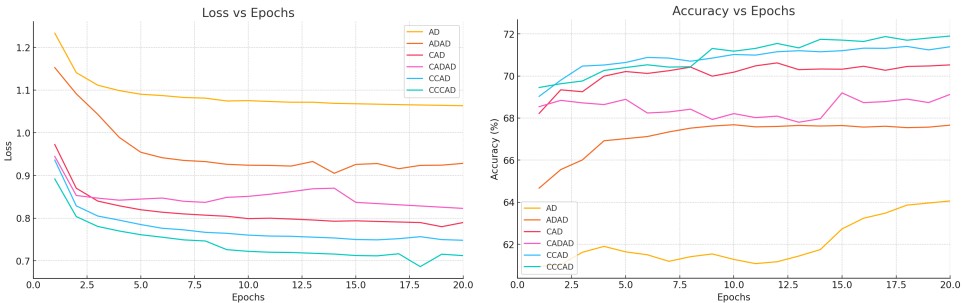

Figure 9: Example loss and accuracy curves of models with different combinations of scaling components trained on 100% of data. (C: convolution module, A: spatial MSA module, E: spectral SDM module, M: one additional set of A and E)

Table 4: Accuracy values of different model combinations under different training data size.

| Model Combination | 10% Data (%) | 20% Data (%) | 50% Data (%) | 100% Data (%) |
|---|---|---|---|---|
| C | 58.2012 | 60.7021 | 63.2015 | 65.2004 |
| CC | 59.7014 | 62.2013 | 64.7032 | 66.2037 |
| CCC | 61.0031 | 63.5043 | 66.0019 | 67.9534 |
| CCCC | 61.0983 | 63.6041 | 66.1057 | 67.9001 |
| CA | 61.4035 | 64.0042 | 67.0025 | 67.0543 |
| CCA | 62.9004 | 65.5053 | 68.5031 | 68.6024 |
| CAA | 63.8032 | 66.4031 | 69.2039 | 69.7528 |
| CD | 60.9023 | 63.5032 | 66.2027 | 68.7539 |
| CDD | 61.0025 | 63.6038 | 66.3029 | 68.6507 |
| CDA | 64.5028 | 66.5041 | 69.0032 | 69.6537 |
| CAD | 63.5041 | 65.5038 | 68.0012 | 70.7543 |
| CADAD | 61.3005 | 63.3034 | 66.0019 | 68.5502 |
| CCAD | 66.2032 | 68.3039 | 70.8024 | 71.5038 |
| CCADAD | 67.5039 | 69.6031 | 72.1035 | 72.8001 |
| CCDA | 65.1008 | 67.1035 | 70.2034 | 70.6532 |
| CCCAD | 66.7025 | 68.8034 | 71.4039 | 71.7635 |
| AD | 61.8004 | 64.3032 | 66.8007 | 64.3449 |
| ADAD | 64.0045 | 66.5034 | 69.0031 | 67.3903 |

As shown in Table 4 and Figure 9, we can summarize the funtions and scaling behaviors of these specific module as follow:

The table presents various model combinations using convolutional blocks (C), multi-head self-attention blocks (A), and spectral dependency modules (D). The performance reflects the trends where A blocks (multi-head self-attention) contribute slightly more significantly to accuracy than D blocks (spectral dependency).

Adding more convolutional blocks steadily improves model performance in the beginning, but the benefit diminishes when convolutional blocks number is high. The base C model achieves an accuracy of 65.20% on 100% of the data. Adding more convolutional blocks improves accuracy, with CC reaching 66.20% and CCC achieving 67.95%. However, the improvement from CCC to CCCC is marginal, showing only a 0.05% increase at 100% data, indicating diminishing returns after three convolutional layers.

The addition of attention blocks (A) brings a substantial boost to model accuracy. For instance, the CA model improves over C by nearly 2%, reaching 67.05% accuracy at 100% data. The improvement is even more significant with deeper combinations like CCA and CAA, both of which surpass CCC. Notably, adding more than one A block, such as in CAA, leads to marginal performance increase compared to CA, but the overall impact of attention remains significant. The best-performing models often include attention blocks, confirming that attention mechanisms contribute strongly to model accuracy.

While D blocks (spectral dependency modules) provide some improvement, and they are significant when combined with attention blocks. For example, the CD model reaches 68.75%, a slight increase over C, but it still lags behind CA. Combinations of A and D, such as CDA and CAD, show significant improvements, with CAD reaching 70.75% on 100% data, highlighting the synergy between these two mechanisms. Considering overall performance, CCAD, CCCAD, CCADAD are at the highest level.

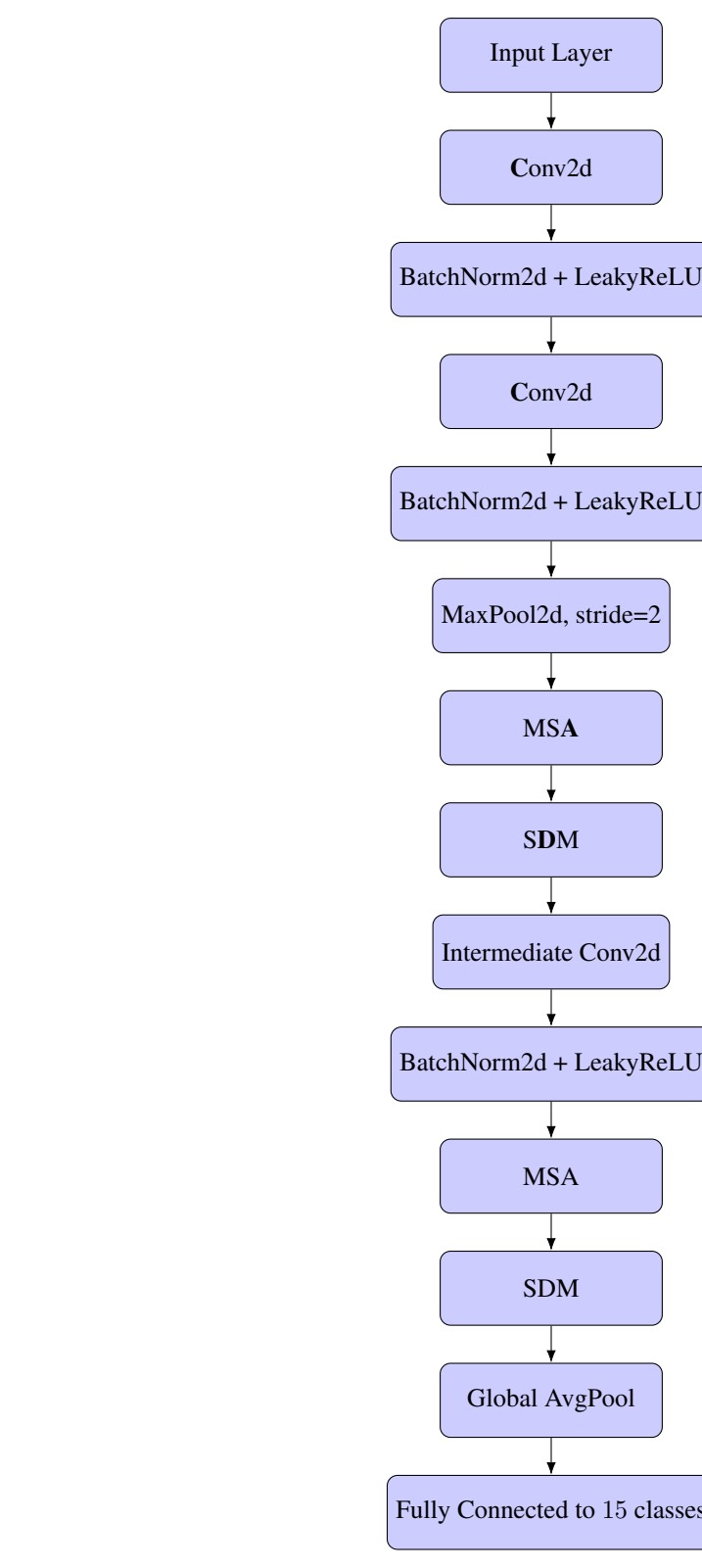

Figure 10: Example architecture of the CC+AD neural network. When MSA and SDM are added to the network, there are always a pair of MSA+SDM models in the model being scaled up at the same time, one placed before intermediate Conv2D, one placed after it.

