# OpenReview forum: "How big does your neural network have to be?: A Scaling Law Study in Multi-Spectral Remote Sensing"
_ICLR.cc/2025/Conference — ICLR 2025 Conference Withdrawn Submission_

### Official Review · Reviewer_mP4r · 2024-10-28

**Soundness:** 1
**Presentation:** 2
**Contribution:** 1
**Rating:** 1
**Confidence:** 4

**Summary:**

The paper proposes a methodology to study the scaling law behaviors of Deep Learning models for remote sensing. In particular, the authors study the scaling behavior of different SOTA architectures on one segmentation dataset. In addition, the authors also propose an adaption of the common attention mechanism, specifically to handle multi-spectral data.

**Strengths:**

The research question is interesting, and relevant for the field of Remote Sensing, as large amounts of data are available and foundation models have significant promises in this area. If scaling laws similar to other domains like LLM can be demonstrated for model architectures, it would support the idea of larger model sizes in this domain.

**Weaknesses:**

The methodology description and experiment setup is missing several relevant details (listed under "Questions"). In addition, the methodology has a series of weaknesses as outlined below that only allow for partial or specific statements of scaling laws. This contrasts the author's tendency to make quiet general statements about model behavior that are invalid under the experiment setup.

Methodological weaknesses:
- In line 160 you state that the labels of LoveDA and BigEarthNet are unreliable. While that might be true, what supports your choice of the BVM dataset. From looking at the BVM publication, I cannot tell for example, whether these labels were manually annotated or automatically generated like it often is the case in EO.
- A main weakness is the choice of just a single recently published dataset on which the authors base all of their conclusion for a subselection of models. In your conclusion you state that "In this paper, we have explored the scaling laws governing neural networks applied to large-scale, multi-spectral remote sensing tasks", but you have only explored a single task on a single dataset. Even within EO segmentation, many specific and relevant tasks exist, like flood segmentation, land-cover classification, crop-type mapping etc. While you recognize this limitation in the subsequent paragraph, it stand in contrast to the general statements you make throughout the paper. For example in line 353, you state that "the MS-Swin" models offer additional advantages, making them a better choice for multi-spectral segmentation tasks", which as a general conclusion does not follow your experiment setup
- Another main weakness is the evaluation scheme, mainly that you only employ Overall Accuracy as a metric for segmentation, and moreover, base all your conclusions on this metric. You state in line 284 that it is a "global measure of how well the model correctly classifies pixels across classes". But this is not necessarily true. The issue here is that for segmentation tasks with high class imbalance like also present in BVM, just predicting a background class or an overly present class will often result in high scores, but does not in itself necessitate a skillfull image segmentation model. Other metrics, like Intersection-over-Union (IoU) or F1 score also need to be considered.
- I am unfortunately confused by your "scaling efficiency metrc", as I cannot find a clear definition of how the individual factors G, P, and C are computed. Additionally, I am not sure where the motivation for this metric comes from, and would be interested in a discussion about its edge cases, where it is defined, for example if G=PxC, etc.
- your introduction of a Spectral Dependency Module is not going into detail about relevant prior research in this domain. Ideas about this have been discussed for example [here](https://ieeexplore.ieee.org/document/8851395)

**Questions:**

Questions about methodology:
- what is the batch size used to train the models? I would assume that it has to be different across models, because of memory constraints, which also has an effect on the training dynamics, of which you include a discussion in your analysis
- how do you choose hyperparameters per experiment setup?
- why do the two resnet model of size 1550 and 2800 have the smallest loss, but significantly lower accuracy than other models? are they overfitting the training data
- in Table 1, there is a "medium models" category (50M to 100M) but the ResNet version entry below shows 230M. Similarly for "large models" there is a 1550M model in the 150M to 300M parameter category
- why does the Unet++ entry for scaling have a N/A entry?
- what loss function are you using for the segmentation task? Cross-entropy loss, Dice loss? Many different options exist in the literature
- how are you adapting the resnet architecture for segmentation task? Is the resnet an encoder for a Unet architecture, or how do you obtain an image representation as the model output?
- in line 141 you state that "few have studied the scaling properties ..., especially in the remote sensing field", however you do not provide any citations. If there are a "few" I would be interested in knowing which ones they are.
- In line 403 you state that transformers are more effective than convolutional architectures when it comes to increasing data and training duration, however, your experimental setup is too limited for that conclusion. For example take the paper by [Woo et al. 2023](https://arxiv.org/pdf/2301.00808) that scales the ConvNext architecture

Other comments:
- in line 278, I think there is a typo, where you mean to say gridding
- in lines 376-378, and 393 to 397 you seem to make the same statement twice? Or could you elaborate between the differences.

---

> ### Author Response · Authors · 2024-11-25
>
> Dear reviewer,
>
> Thank you for your comment, we will include more datasets and address this problem in future editions.
>
> Best,

---

### Official Review · Reviewer_79Kq · 2024-11-01

**Soundness:** 2
**Presentation:** 2
**Contribution:** 2
**Rating:** 1
**Confidence:** 5

**Summary:**

This paper presents an analysis of how different model architectures perform on the Biological Valuation Map of Flanders across different numbers of parameters of each architecture. The authors present a modified Swin transformer architecture designed to learn spectral features through a Spectral Dependency Module which performs attention along the spectral dimension, aggregating all pixels into a single patch embedding. Experiments show that Swin transformer architectures typically have higher accuracy across all scales and have greater scaling efficiency.

**Strengths:**

- The authors aim to answer an important question about scaling laws for remote sensing data, i.e., how we should size model architectures given a particular dataset

**Weaknesses:**

- The scope of the question the authors claim to answer in the beginning of the paper is mismatched with the scope of their experiments. The general claims made in the paper about scaling laws and architecture preferences are all based on a single dataset, which is large in terms of number of pixels but limited in terms of geographic/task diversity. The claims can only be made about that dataset, not generalized to multispectral remote sensing datasets as a whole.

- The scope of datasets included in Figure 1 is very limited. There are many popular datasets not included in this, such as EuroSat and Functional Map of the World, to name a couple.

- The writing quality is poor throughout the paper. There are many vague or poorly worded statements that obscure the authors' intended meaning, especially in Section 2.1.

- The presentation of the results in Table 1 should be improved. Some rows seem like they are not in the appropriate category, e.g., ResNet-230M is in the 50M-100M parameter category and ResNet-1550M is in the 150M-300M parameter category. The results are given for one model run so we cannot assess the variation due to randomness in the results (multiple runs should be performed with the mean and standard deviation reported).

- The authors claim the results for the MS-Swin models (using their proposed SDM module) are significantly better than without. However, these differences are likely not significant - for most comparisons they are less than 0.1% higher. Thus it is misleading to claim that their proposed model "achieves superior accuracy and scaling efficiency".

- The authors point out some of these limitations in Section 5, which I commend them for. However, the limitations are too great to accept this paper.

**Questions:**

I do not have questions that would change my opinion, since I feel the weaknesses are with the scope of the experiments and the interpretation of the results.

---

> ### Author Response · Authors · 2024-11-25
>
> Dear reviewer,
>
> Thank you for your comment, we will apply the model to other datasets and address the problems you mentioned.
>
> Best,

---

### Official Review · Reviewer_27tX · 2024-11-03

**Soundness:** 2
**Presentation:** 1
**Contribution:** 2
**Rating:** 3
**Confidence:** 4

**Summary:**

This work presents an analysis on the relationship between size of model architecture and performance for multiple architectures and multispectral imagery datasets. Authors introduced a modification of the spatial attention head of the transformer architecture to apply the attention mechanism over the spectral dimension of the multispectral imagery which is refered to as Spectral Dependency Module (SDM). Modifying the Swin Transformer with the SDM improved model performance over the Swin transformer on the Biological Valuation Map (BVM) dataset.

**Strengths:**

* The Spectral Dependency Module (SDM) seems to be effective at leveraging the spectral data in multispectral imagery for the tasks studied.
* The MS-swin transformer outperforms the standard swin transformer on the BVM dataset

**Weaknesses:**

* The analysis is conducted on a very reduced set of datasets from local areas which limits any findings from this work. Expanding the analysis to other datasets will benefit the work.
* The work is framed as a study on the scaling laws of neural networks on multispectral images however it mostly focus on analysing the performance of the proposed MS-swin tranformer and do not consider other transformer based architectures.
* It is difficult to draw any conclusion from the analysis presented on the scaling laws of neural networks.
* The paper needs some work to make it more cohesive.

**Questions:**

* Can you motivate better why the scaling laws of multispectral images would be different to to natural images?
* Text and legends for most figures are difficult to read. Consider making text bigger.
* The MS-swin transformer can be a strong contribution and can be the focus of an article. However, it by itself does not answer the analysis framed on scaling laws of neural network in remote sensing.

---

> ### Author Response · Authors · 2024-11-25
>
> Dear reviewer,
>
> Thank you for your precious comment, we will try to extend the experiment to other datasets and revise in the future.
>
> Best,

---

### Official Review · Reviewer_Z7UX · 2024-11-03

**Soundness:** 2
**Presentation:** 1
**Contribution:** 2
**Rating:** 3
**Confidence:** 4

**Summary:**

The paper presents a Module for handling multi-spectral data in deep learning models (SDM). This methodological proposition is embedded in a scalability study in which differently sized model architectures (Unet++, ResNet, SwinTransformers) are evaluated on Sentinel-2 images from Flanders (BVM dataset of Li et al.). Notably the DSM module is only implemented in the SwinTransformers and not the other Architectures.

Overall, the paper lacks a certain focus on a single research question and contribution and jointly investigates two topics without going in-depth in either of them: First, a scalability study (Title & Abstract) would be interesting in general, as certainly some insights can be gained from training large-scale neural networks in remote sensing Terrabytes of remote sensing data in an experimental setting analogue to training large language models (see, for instance, Kaplan et al., 2020). However, the paper falls short in this direction, as all experiments are exclusively conducted on a particularly specific and not well-used Sentinel-2 land cover map where the actual dataset size does not seem to be mentioned in the paper. Also, the final conclusions in this regard are fairly limited, as larger models perform better (P9 L463), which is somewhat expected.

The second contribution in proposing an attention mechanism to handle multi-spectral data is interesting, but also not investigated further beyond an ablation study on a single model architecture (SWIN). Here, several other papers have investigated how to dynamically input a varying number of bands into neural networks through tokenization and spectral-wise attention (see Tseng et al., 2023 and Nguyen et al., 2023, for instance), which seem methodologically (Section 3.1) very close to the proposed attention mechanism.

Finally, the presentation and results are not very well prepared. The figures are too small, and the labels are not readable.


Kaplan, Jared, Sam McCandlish, Tom Henighan, Tom B. Brown, Benjamin Chess, Rewon Child, Scott Gray, Alec Radford, Jeffrey Wu, and Dario Amodei. "Scaling laws for neural language models." arXiv preprint arXiv:2001.08361 (2020).

Nguyen, T., Brandstetter, J., Kapoor, A., Gupta, J. K., & Grover, A. (2023). ClimaX: A foundation model for weather and climate. arXiv preprint arXiv:2301.10343.

Tseng, G., Cartuyvels, R., Zvonkov, I., Purohit, M., Rolnick, D., & Kerner, H. (2023). Lightweight, pre-trained transformers for remote sensing timeseries. arXiv preprint arXiv:2304.14065.

**Strengths:**

* Interesting problem setups (investigating scaling laws and the peculiarities of multi-spectral data)

**Weaknesses:**

* lack of focus on the underlying research question: Is this a scalability study or the proposition of a spectral module?
* Results are not very informative (large models perform better on a large enough dataset). What are more fine-grained takeaways? (Maybe also beyond the remote sensing context)

**Questions:**

See questions in weaknesses.

---

> ### Author Response · Authors · 2024-11-25
>
> Dear reviewer,
>
> We will address the problems and try to focus more on one research problem in future revisions.
>
> Best,

---

### Official Review · Reviewer_rdSx · 2024-11-06

**Soundness:** 2
**Presentation:** 3
**Contribution:** 2
**Rating:** 3
**Confidence:** 4

**Summary:**

This experimental paper investigates how scaling model and dataset sizes impacts segmentation accuracy on multispectral remote sensing data. The authors evaluate Swin Transformer models, both with and without their proposed Spectral Dependency Module (SDM), alongside ResNet architectures. The SDM applies attention across the channel dimension, rather than on patch tokens, to capture spectral dependencies. Experiments demonstrate that increasing both model and dataset size leads to notable accuracy improvements.

**Strengths:**

The paper is easy to read and the experiments studying scaling laws for multispectral images could be promising for remote sensing community. Also, the study shows that Swin Transformer based architectures are better suited for their tasks.

**Weaknesses:**

While the study of scaling laws for multispectral images may hold interest for the remote sensing community, it may have limited relevance for a broader ICLR readership. The general finding—that increasing model and dataset size improves accuracy—is well-established, and Transformer-based architectures are already known to perform well under such conditions. The primary novelty claimed by the authors is the Spectral Dependency Module (SDM). However, I have reservations about its actual impact on performance. Specifically, I question whether capturing long-range dependencies through attention is necessary in this context, as a linear layer could potentially offer similar computational strength. Furthermore, Figure 4 suggests that the best model is the one without SDM, which raises concerns about SDM’s effectiveness. There is also an inconsistency between Table 1 and Figure 4 that requires clarification. Additionally, the significance of SDM's reported improvements in Table 1 is uncertain, as no confidence intervals, standard deviations, or other statistical measures are provided.

Other minor remarks:

- The abstract mentions that convolutional models typically only accept three channels, but they can be adapted to handle a larger number of channels.

- The accuracy comparisons in Figure 1 are challenging to interpret, and comparing across different datasets may not be entirely fair. Separate plots showing accuracy versus dataset size for a single data distribution and accuracy versus model size would make the findings more digestible.

- Symbols such as $N_p$ and $P$ should be defined for clarity.

- Distinguishing between models in Figure 7 is difficult; improved visual differentiation is recommended.

- The use of boldface in Table 2 (Appendix) could be misleading, as different datasets are compared.

**Questions:**

1- In lines 200-205 the authors claim that _by computing attention weights based on spectral dependencies, the SDM allows for a fine-grained representation of spectral relationships_. Could the authors elaborate on how the SDM achieves this fine-grained representation, specifically in terms of its design and intended impact on model performance?

2- In line 246, it is mentioned that $\mathbf{O}$ is averaged over all patches from the final representation used for downstream tasks.
Is $\mathbf{O}$ directly used in the segmentation task, and if so, how does it contribute to distinguishing pixels from different patches? Additionally, how is the output of SDM integrated with subsequent layers in the network?

3- What does "stage 1" refer to in line 269?

---

> ### Author Response · Authors · 2024-11-25
>
> Dear reviewer,
>
> Thank you for your precious comment, we take your comment seriously and will revise the paper accordingly in the future.
>
> Best,

---

### Note · Authors · 2024-11-25

I have read and agree with the venue's withdrawal policy on behalf of myself and my co-authors.